# Influence of Terrain Factors on Urban Pluvial Flooding Characteristics: A Case Study of a Small Watershed in Guangzhou, China

Xuelian Zhang [1,2], Aiqing Kang [2,*], Mao Ye [2], Qingxin Song [3], Xiaohui Lei [2] and Hao Wang [2]

1   School of Civil Engineering, Central South University, Changsha 410075, China; qingan@csu.edu.cn
2   State Key Laboratory of Simulation and Regulation of Water Cycle in River Basin, China Institute of Water Resources and Hydropower Research, Beijing 100038, China; yemao@iwhr.com (M.Y.); lxh@iwhr.com (X.L.); wanghao@iwhr.com (H.W.)
3   School of Water Conservancy and Hydroelectric Power, Hebei Engineering University, Handan 056038, China; songqingxin1994@163.com
*   Correspondence: kaqing@iwhr.com; Tel.: +86-1352-095-7120

**Abstract:** Urban roads in China, particularly low-lying areas such as underpasses, tunnels, and culverts, are highly vulnerable to the dangers of urban pluvial flooding. We used spatial interpolation methods and limited measured data to assign elevation values to the road surface. The road network was divided into tiny squares, enabling us to calculate each square's elevation, slope, and curvature. Statistical analysis was then employed to evaluate the impact of terrain on flood characteristics in urban road systems. Our analysis reveals a strong spatial correspondence between the distribution of flood-prone points and the curvature parameters of the terrain. The spatial coincidence rate can reach 100% when an appropriate sampling scale is chosen. The presence of depressions is necessary but insufficient for forming flood-prone points. In lowland/gentle slope (LL/GS) areas with higher drainage pressure, we observe a significant negative correlation between flood-prone points and terrain curvature (Spearman's r = 0.205, $p < 0.01$). However, in highland/steep slope (HL/SS) areas, we find no significant correlation between them. Notably, terrain matters, but effective drainage is more influential in flood-prone areas. The maximum flood depth (MFD), submerged area, and ponding volume during urban pluvial flooding are constrained by depression topography, while the characteristics of the upstream catchment area also play a role in determining the MFD and flood peak lag time(FPLT). Larger upstream catchment areas and longer flow paths normally result in greater MFD and longer emergency response times/FPLT. Additionally, a higher flow path gradient will directly contribute to an increased flood risk (greater MFD and shorter FPLT). These findings have important implications for flood risk identification and the development of effective flood mitigation strategies.

**Keywords:** urban pluvial flooding; flood-prone point; terrain; flood characteristics

## 1. Introduction

Urban pluvial flooding is a natural disaster that has received much attention in China in recent years, resulting in significant loss of life and property [1–3]. Flooding happens when rainwater builds up or flows locally on the ground before entering the sewer drainage system, or when the sewer drainage system becomes overwhelmed and cannot handle additional rainwater [4,5]. There is currently limited knowledge of the occurrence and expanding patterns of urban pluvial flooding compared to the reasonably mature research on river and coastal flooding [5,6]. Urban pluvial flooding is believed to be highly related to rainfall patterns, urban surface characteristics (including terrain and land feature variables), and drainage system performance [7–10]. However, the specific mechanisms of these factors remain highly uncertain [5,6,11–13]. As a result, researchers have devoted

considerable efforts to analyzing the triggering factors of urban pluvial flooding, to identify the mechanisms that cause this disaster and take targeted measures to reduce the risk of urban pluvial flooding [7,11,14–17].

In contrast to undeveloped rural areas, urban regions typically incorporate comprehensive drainage design during the early stages of development. The current urban drainage system typically comprises two main components: the minor drainage system (sewer network) designed for storm events with return periods of up to 10 years, and the major drainage system (overland flow) engineered to handle larger flows from storm events with return periods up to 100 years [18–22]. Theoretically, during storm events within the drainage capacity of minor drainage systems, the main drainage systems, including roads, drainage ditches, and rivers, should be immune to flood hazards [11,18,20,21]. However, in recent years, there has been a significant increase in flood risk on urban road networks [23–26], even when exposed to rainfall events normally managed by minor drainage systems [24]. This escalation can be attributed to factors such as inadequate maintenance practices [27], outdated drainage infrastructure unable to cope with urbanization [27], and neglect of hydraulic efficiency in inlet design [28–30]. The trapping of people in vehicles on flooded roads has led to fatalities in numerous urban flood events [31,32]. As a result, assessing flood risk in urban road networks has become a critical aspect of studying urban pluvial flooding. This has led to an increased focus on assessing flood risk in urban road networks in recent years [33–38].

Terrain plays a fundamental role in the dynamics of hydrological systems [9,39–41], exerting significant control over the distribution and infiltration of surface runoff and regulating its depth and spatial distribution [11,42–45]. Consequently, it has a crucial influence on the characteristics of urban floods [46–51]. Previous research on the impact of terrain on runoff or flood events has primarily relied on indirect indicators, such as surface roughness at scales exceeding decimeters [52,53] or connectivity [38,49–51]. Alternatively, studies have focused on comparing elevation data from different sources with varying degrees of accuracy to evaluate the performance of hydrological models. However, there is a noticeable lack of research specifically investigating the direct relationship between terrain parameters and flood characteristics. This scarcity can be attributed to the challenges of obtaining accurate terrain data for urban areas. Digital elevation models (DEMs) or surveying techniques are commonly used to derive urban topographic data [54]. However, DEMs in urban areas are prone to localized distortions due to artificial structures like buildings and bridges [54]. Furthermore, conducting comprehensive topographic surveys on an urban scale proves impractical due to the extensive resources required. To address these limitations, this study employs spatial interpolation methods to extrapolate elevation values from a limited set of measured points to the entire road surface [54]. Inspired by approaches used in ecological research to investigate the influence of terrain on species richness [55–57], statistical analysis techniques are employed to examine the associations between commonly observed terrain parameters (elevation, slope, curvature) and flood characteristics on urban road surfaces. The primary objective is to quantitatively elucidate the impact of terrain on the characteristics of floods in urban road networks.

Our research addresses the following questions: (1) Is there a relationship between the distribution of flood-prone points and terrain factors, and if so, how do terrain parameters influence the distribution of flood-prone points? (2) Have terrain factors affected the occurrence and development process of surface flooding, and if so, what are the specific effects? We hypothesize that (1) the degree of terrain depression is one of the primary factors affecting the distribution of flood-prone points and should not be overlooked while analyzing the urban pluvial flooding risk; (2) the influence of various terrain parameters on the distribution of flood-prone points differs depending on the type of terrain; and (3) terrain factors are essential for determining the maximum flood depth(MFD), submerged area, and ponding volume of urban pluvial flooding at specific flood-prone points. They significantly contribute to assessing the systemic risk associated with urban pluvial flooding in a specific area. This research aims to enhance our understanding of the mechanisms underlying

urban pluvial flooding and provide a robust scientific basis for developing comprehensive plans to manage emergencies and prevent urban flooding.

## 2. Materials and Methods

### 2.1. Study Area

This study was conducted in a small watershed (which belongs to the Pearl River system and has a drainage area of approximately 15.82 km$^2$) in Guangzhou, a city in southern China. The Little River watershed is a high-risk location for urban pluvial flooding since highlands characterize the higher portions, and the lower levels are tidal and prone to flooding. The climate of this study area is classified as maritime subtropical monsoon climate, with average annual precipitation and evaporation of 1600–1900 mm and 1700–1800 mm, respectively. This study area has extremely varied terrain, with an elevation difference of about 40 m and slopes primarily ranging from 0° to 20°, all of which are urban built-up areas. Most rainfall occurs from April to September, accounting for over 70% of the annual rainfall, with average annual temperature ranging from 20 to 28 °C.

### 2.2. Road Investigation and Parameter Calculation

#### 2.2.1. Terrain Parameters of Each Square

The specified study area has been divided into numerous tiny squares, as illustrated in Figure 1, to extract a comprehensive set of topographical data, four distinct scales were used, including 27 m × 27 m, 54 m × 54 m, 81 m × 81 m, and 108 m × 108 m, to minimize any potential mistakes resulting from different sampling scales.

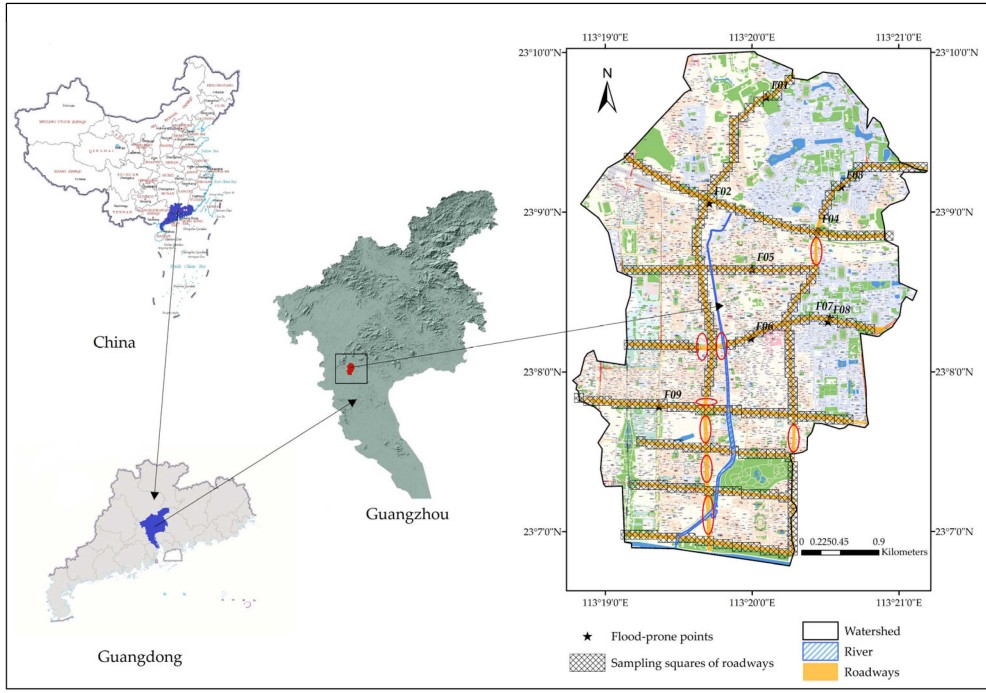

**Figure 1.** Location and sample squares of the study area.

The specific sampling method is as follows: first, ArcGIS10.5 was used to split the study's area into grids of 1 m × 1 m, 2 m × 2 m, 3 m × 3 m, and 4 m × 4 m. By combining these grids, sample squares with sizes of 27 m × 27 m, 54 m × 54 m, 81 m × 81 m, and 108 m × 108 m were made for the roadways in the research region. Each grid's core points were saved for later study. Due to challenges measuring surface elevation and the lack of floodwater accumulation, the bridge and other construction locations inside the red ellipse were excluded from the sampling range, as indicated in Figure 1. The next step was to determine the topographical attributes for each square, including elevation (which utilizes relative values rather than height numbers due to limitations of the data provider),

curvature, and slope. We adopted the principle of proximity and assigned the measured elevation data to the center point of each small grid using ArcGIS10.5. The 729 elevation-valued points within each square that result from this process were used to determine the terrain parameters within each square. Each sample square's elevation is the average of the elevations of all points within the square. Finally, using the slope and curvature calculation tools offered by ArcGIS software, these squares' slopes, and curvatures were determined using the center points whose elevation values are provided above.

### 2.2.2. The Mark of Each Square

Each square was marked with a binary value to aid in analysis. A value of "1" indicated overlap with areas of historical flood incidence recorded by the Guangzhou Water Affairs Bureau, while a value of "0" indicated no overlap.

### 2.2.3. Morphology Parameters and Catchment Parameters for Each Depression

Urban roads are designed with a nearly consistent height along the same cross-section to accommodate the needs of both vehicles and pedestrians [58]. In low-lying places where flooding may occur, this analysis focuses primarily on the longitudinal profile of the road and the maximum probable inundation length (B) and water depth (H). The criterion used to assess these areas' form properties is the ratio of B/H. There is currently no consensus on the definitive approach to characterizing the morphology of depressions. The width/depth ratio is a critical parameter in assessing river channel morphology. In the context of urban flood events, urban roads frequently assume partial functions of urban rivers. Hence, in this study, we have chosen to employ the ratio of B/H to evaluate the morphology of depressions [59–61].

For the catchment parameter of the depression, we primarily focus on three key factors: catchment area (A), flow path length (L), and flow path gradient (G), and these parameters can be obtained using a digital elevation model [62,63].

### 2.2.4. Characteristic Parameters of Flood

This study identifies the MFD and flood peak lag time (FPLT) as the key parameters of urban pluvial flooding. MFD refers to the water level corresponding to the peak flow during an urban flood event [64,65], while FPLT represents the time interval between the occurrence of the maximum flood depth and the maximum rainfall intensity [66,67]. MFD and FPLT are key indicators in flood forecasting and risk analysis.

### 2.3. Sources of Data

The data on the distribution of flood risk points were obtained from the flood risk point data released by the Guangzhou Municipal Emergency Management Bureau in May 2022 [68]. The Guangzhou Water Affairs Bureau has provided roadway elevation data, measured flood process data for flood-prone points, and precipitation data. There exist nine historical flood-prone points within the study area. These points are identified by their corresponding numbers, namely F01, F02, F03, F04, F05, F06, F07, F08, and F09, as can be seen in Figure 1.

Highly sensitive water level monitoring devices were installed at flood-prone locations in this study. It should be noted that passing vehicles on the road may cause abnormal fluctuations in water level readings. To examine the impact of topography on the MFD, flood events with a maximum depth recording of less than 10 cm were excluded from this study. Based on this criterion, data from five flood-prone locations recorded on 23 April and 7 June 2022 were selected for further analysis. When investigating the influence of topography on the lag time of flood peaks, it is worth noting that the data selection criteria can be relaxed. Consequently, there is an opportunity to include additional ponding data from 26 March 2022 in the research analysis. This inclusion is expected to refine and enhance the precision of the research conclusions.

### 2.4. Statistical Analysis Method

SPSS 22.0 software was utilized for data analysis, encompassing the following procedures: (1) The mean ± standard deviation was employed to represent the terrain parameters for each terrain classification, while one-way analysis of variance (ANOVA) was employed to examine differences among the various terrain categories. (2) Spearman's correlation test was applied to assess the association between the distribution of flood-prone locations and terrain parameters. Spearman's rank correlation coefficient, devised by the British statistician Spearman, was used to evaluate the correlation between categorical and ordinal variables. (3) Pearson's correlation test was used to evaluate the relationship between urban flood characteristic parameters and the morphological parameters of depressions and watershed regions. Pearson's correlation coefficient is suitable for exploring correlations among continuous variables. (4) Significance levels were set at $p < 0.05$ to denote statistically significant differences and $p < 0.001$ to indicate highly significant differences. (5) The letters "a" and "b" were used to indicate statistically significant differences between variables, with the same letter denoting non-significant differences and different letters signifying significant differences [69–71].

Referring to the method of micro terrain classification in ecology, R4.2.1 was used to conduct C-means fuzzy cluster analysis for the terrain parameters of each sample in the study area [72,73].

### 3. Results

#### 3.1. Impact of Terrain on the Distribution of Urban Pluvial Flooding

##### 3.1.1. Spatial Correspondence between the Distribution of Flood-Prone Points and Terrain Characteristics

A spatial superimposition analysis of the distribution of flood-prone spots and the terrain parameters was conducted to investigate the relationship between the two. As shown in Figure 2, the total number of historical flood-prone points in the study area is nine. Under the four scales of 27 m × 27 m, 54 m × 54 m, 81 m × 81 m, and 108 m × 108 m, the number of flood-prone points that coincide with the quadrats with curvature < 0 in the terrain parameters are 6, 9, 8, and 8, respectively. The spatial coincidence rate can reach a maximum of 100%, indicating a strong spatial correspondence between the distribution of urban flood-prone points and the micro-topographic pattern. The sampling grid size also influences the research findings: when the sampling scale is 54 m × 54 m, the coincidence ratio between flood-prone points and terrain depression parameters is higher, and when it is larger or smaller than this scale, the degree of coincidence decreases. The road width in the study area is generally between 10 and 60 m, and it seems that the closer the side length of the sampling grid is to the road width, the more realistic the calculated terrain parameters are.

##### 3.1.2. Correlation between Distribution of Flood-Prone Points and Terrain Parameters

The results presented in Table 1 provide clear evidence of a significant correlation between terrain curvature and the distribution of flood-prone points across all sampling scales. Corresponding to this, no significant correlation was found between elevation and slope. Specifically, our findings indicate that at scales of 27 m × 27 m and 54 m × 54 m, there was a significant negative correlation between terrain curvature and the distribution of flood-prone points ($p < 0.05$), while at other scales, the correlation was even more significant ($p < 0.01$). Greater surface concavity corresponds to a higher risk of stagnant floodwaters, while elevation and slope show no significant influence on the distribution of flood-prone areas. These results highlight the crucial influence of terrain curvature on the distribution of urban flood-prone points and suggest the need for further investigation into this relationship.

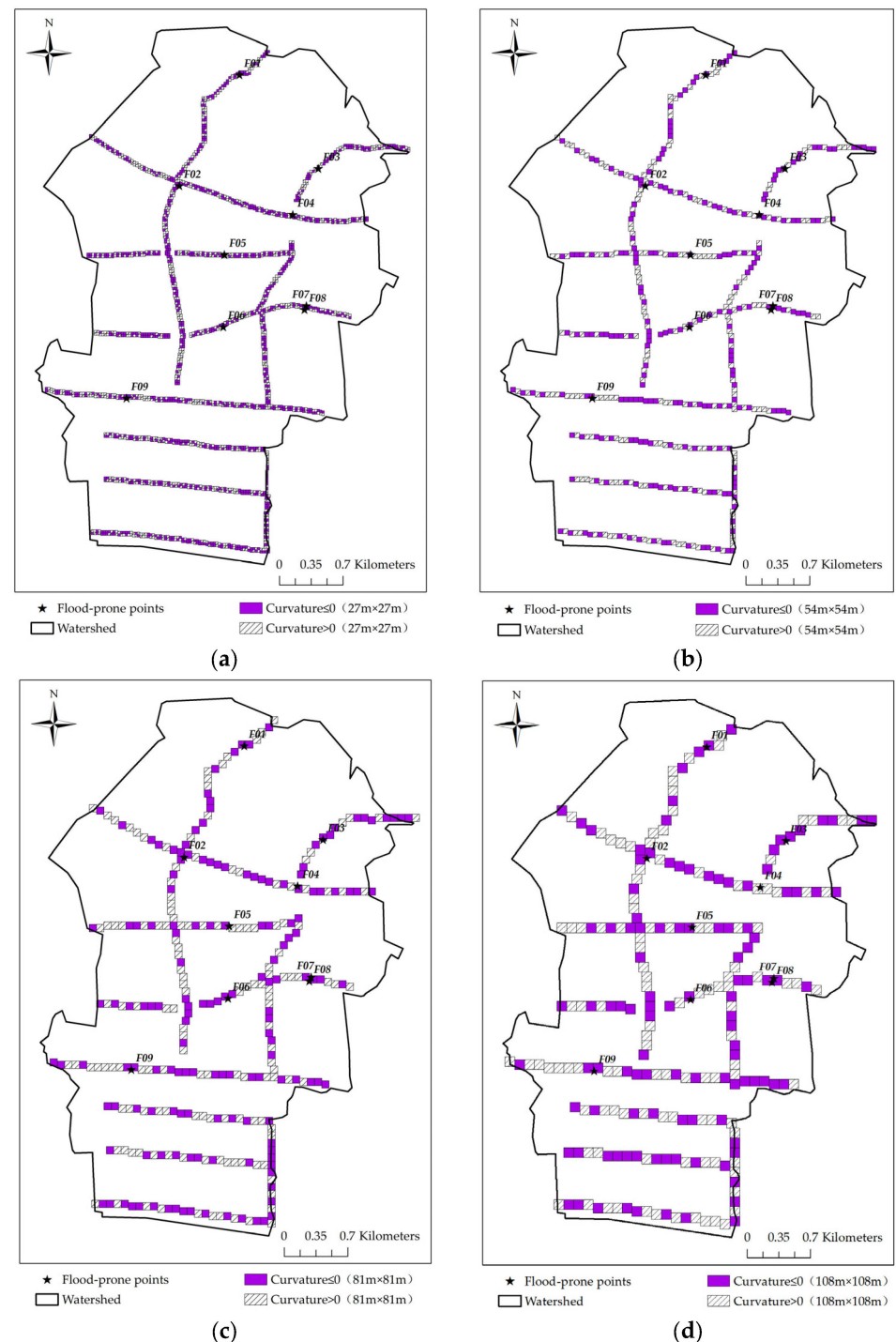

**Figure 2.** Spatial correspondence between the surface depression and the flood-prone points at different scales. (**a**) 27 m × 27 m; (**b**) 54 m × 54 m; (**c**) 81 m × 81 m; (**d**) 108 m × 108 m.

**Table 1.** Correlation between distribution of flood-prone points and terrain parameters.

| Terrain Parameter | Scale (m × m) | Elevation (m) | Curvature | Slope (°) |
|---|---|---|---|---|
| correlation coefficient | 27 × 27 | 0.028 | −0.054 * | 0.026 |
| | 54 × 54 | 0.058 | −0.182 ** | 0.084 |
| | 81 × 81 | 0.073 | −0.145 * | 0.077 |
| | 108 × 108 | 0.087 | −0.219 ** | 0.126 |

Note(s): [1] *: $p < 0.05$; [2] **: $p < 0.01$.

### 3.1.3. Terrain Classification and Correlation between the Distribution of Flood-Prone Points and Terrain Parameters under Different Terrain

The study's area has been classified into two distinct terrain types, namely highland/steep slope (HL/SS) areas, and lowland/gentle slope (LL/GS) areas, using three terrain parameters: elevation, slope, and curvature. Table 2 describes each terrain type's characteristics and the number of squares. The results show that at a sampling scale of 27 m × 27 m, significant differences ($p < 0.05$) were observed in the three terrain parameters between HL/SS areas and LL/GS areas. However, at other scales, only elevation and slope demonstrated significant differences ($p < 0.05$), whereas curvature parameters showed no significant differences. This implies that the distribution of surface depressions (curvature < 0) on urban road surfaces within the study area is not influenced by variations in elevation or slope. These depressions appear to be randomly distributed across all areas of the road network.

**Table 2.** Features of two types of terrain and the corresponding number of squares.

| Scale (m × m) | Terrain Parameter | Lowland/Gentle Slope (LL/GS) | Highland/Steep Slope (HL/SS) |
|---|---|---|---|
| 27 × 27 | Total number of squares (TNS) | 1363 | 428 |
| | Mean elevation (m) | $-0.545 \pm 3.457$ a | $13.169 \pm 6.386$ b |
| | Curvature | $0.011 \pm 0.721$ a | $-0.036 \pm 0.295$ b |
| | Slope (°) | $0.485 \pm 0.714$ a | $1.123 \pm 1.484$ b |
| | Number of flood-prone squares (NFPS) | 7 | 2 |
| | NFPS/TNS (%) | 0.51 | 0.47 |
| 54 × 54 | TNS | 350 | 111 |
| | Mean elevation (m) | $-0.437 \pm 3.515$ a | $13.230 \pm 6.797$ b |
| | Curvature | $-0.004 \pm 0.558$ a | $-0.007 \pm 0.163$ a |
| | Slope (°) | $0.489 \pm 0.710$ a | $1.173 \pm 1.296$ b |
| | NFPS | 7 | 2 |
| | NFPS/TNS (%) | 2.00 | 1.80 |
| 81 × 81 | TNS | 223 | 85 |
| | Mean elevation (m) | $-0.309 \pm 3.548$ a | $11.146 \pm 8.701$ b |
| | Curvature | $-0.001 \pm 0.019$ a | $-0.007 \pm 0.116$ a |
| | Slope (°) | $0.388 \pm 0.511$ a | $1.309 \pm 1.175$ b |
| | NFPS | 7 | 2 |
| | NFPS/TNS (%) | 3.14 | 2.35 |
| 108 × 108 | TNS | 161 | 69 |
| | Mean elevation (m) | $-0.3241 \pm 3.496$ a | $10.146 \pm 8.892$ b |
| | Curvature | $-0.000 \pm 0.017$ a | $-0.005 \pm 0.034$ a |
| | Slope (°) | $0.330 \pm 0.382$ a | $1.332 \pm 1.121$ b |
| | NFPS | 6 | 3 |
| | NFPS/TNS (%) | 3.73 | 4.35 |

Note(s): [3] "a" and "b": The letters "a" and "b" were used to indicate statistically significant differences between variables, with the same letter denoting non-significant differences and different letters signifying significant differences $p < 0.05$.

The classification findings are shown in Figure 3, which shows that steep slopes are generally found in regions with higher surface elevations, while gentle slopes are primarily found in regions with lower surface elevations. These findings are consistent with the general rule that terrain in estuary areas tends to be relatively flat. Additionally, at all sizes, there are significantly more squares classed as LL/GS areas than HL/SS areas. Interestingly, there is little difference between the ratio of flood-prone squares to all squares in these two types of terrain. Given the previous finding that the distribution of depressions is not influenced by surface elevation or slope, it can be inferred that the distribution of flood-prone points is also not significantly affected by elevation or slope.

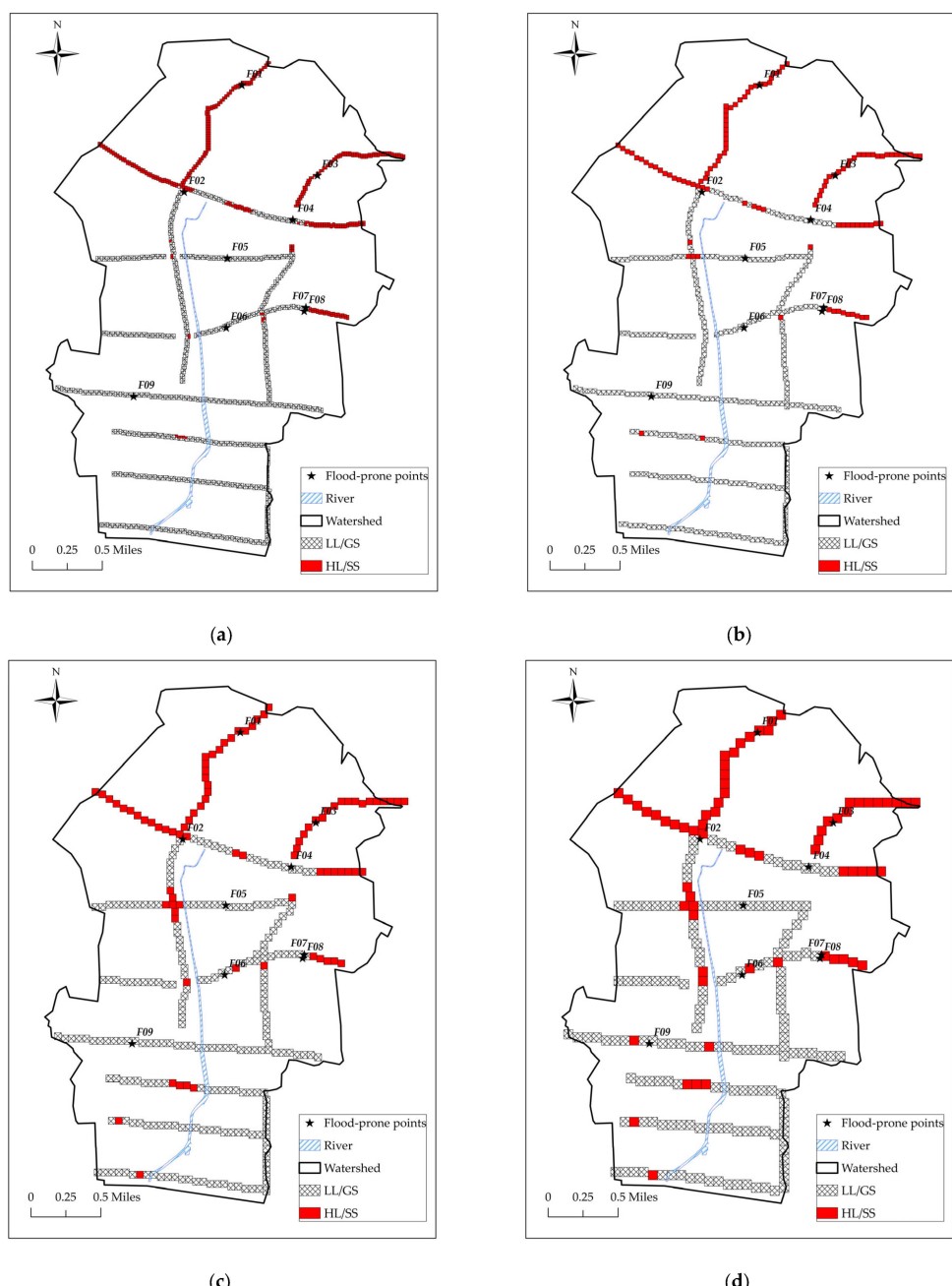

**Figure 3.** Distribution of highland/steep slope (HL/SS) and lowland/gentle slope (LL/GS) squares on different scales. (**a**) 27 m × 27 m; (**b**) 54 m ×54 m; (**c**) 81 m × 81 m; (**d**) 108 m × 108 m.

Table 3 reveals a negative correlation between the distribution of flood-prone points and curvature in the LL/GS areas at all scales, with a highly significant negative correlation ($p < 0.01$) at the sampling scale of 54 m × 54 m and 108 m × 108 m, and a significant negative correlation ($p < 0.05$) at other scales. Regarding slope parameters, there is no significant relationship between the distribution of flood-prone points and slope at the sample scale of 54 m × 54 m, while there is a significant positive correlation at other scales ($p < 0.05$). The distribution of flood-prone points and the elevation in the LL/GS areas does not significantly correlate at any scale. Overall, terrain characteristics clearly influence the distribution of flood-prone points in LL/GS areas.

**Table 3.** Correlation between distribution of flood-prone points and terrain parameters under different types of terrain.

| Terrain Parameter | Scale (m × m) | Terrain Categorization | Elevation | Curvature | Slope |
|---|---|---|---|---|---|
| Correlation coefficient | 27 × 27 | LL/GS | 0.046 | −0.068 * | 0.048 |
| | | HL/SS | 0.030 | −0.012 | −0.046 |
| | 54 × 54 | LL/GS | 0.093 | −0.205 ** | 0.119 * |
| | | HL/SS | 0.057 | −0.112 | −0.030 |
| | 81 × 81 | LL/GS | 0.115 | −0.149 * | 0.170 * |
| | | HL/SS | 0.085 | −0.123 | −0.142 |
| | 108 × 108 | LL/GS | 0.107 | −0.214 ** | 0.201 * |
| | | HL/SS | 0.039 | −0.218 | −0.054 |

Note(s): [4] *: $p < 0.05$; [5] **: $p < 0.01$.

Answer relatively to it, there is no apparent correlation between the distribution of flood-prone points and the three types of terrain parameters in areas with HL/SS at different scales.

### 3.2. Influence of Topography on the Characteristics of Urban Pluvial Flooding

Five flood-prone depressions (F02, F04, F05, F06, and F08) met the research criteria in 2022. The maximum standing water depths for each flood-prone point are 1.73 m, 2.22 m, 0.96 m, 0.75 m, and 2.08 m, respectively, as shown in Figure 4, which shows the vertical section of these depressions. Furthermore, the figure displays the maximum inundation lengths for each point, which are 161 m, 1182 m, 353 m, 547 m, and 297 m, respectively. Finally, by dividing the vertical section area corresponding to each depression by the road width, we can estimate the maximum ponding volume and water level-volume curve of each depression. These results will all serve as reliable foundations for developing flood prevention plans.

Table 4 shows the characteristic parameters of urban pluvial flooding in five flood-prone locations on 23 April 2022, and 7 June 2022. During the two flood incidents, we evaluated and rated the discrepancy in MFD at each flood-prone point. The results show that F02 had the highest water depth variation value, followed by F08, F06, F05, and F04. The MFD at F06 and F05 can only fluctuate within the range of 0 to 0.75 m and 0 to 0.96 m, respectively, due to the depression's maximum depth limit, as a result, the variation of the MFD at F06 and F05 is significantly less than that of F02 and F08. The depression's maximum depth of F04 is reaching levels comparable to F02 and F08. However, the vertical section of the depression shown in Figure 4 reveals that the maximum ponding volume in the depression where F04 is situated is significantly greater than others. This suggests that if the water level rises per unit depth, the depression would need to accumulate more floods. This results in the minor modification of the MFD of the flood-prone point F04 in the two flood events in this study. Notably, the FPLT for each flood event, as observed in the two flood processes at the five flood-prone points, ranging from 0 to 21 min, indicates the rapid onset of urban pluvial flooding. This highlights the limited time available for emergency response.

Figure 5 explores the correlation between the morphology parameter (B/H) of depressions and flood characteristic parameters (MFD) of flood-prone points, while Figure 6 examines the correlation between the depression's catchment parameters (A, L, G) and flood characteristic parameters (MFD and FPLT) of flood-prone points. The result demonstrates that for a single flood event, the MFD of various flood-prone points is positively correlated with the A, L, and G of the depression's catchment but negatively correlated with the B/H of the vertical section of the depression to which each flood-prone point belongs. For FPLT, it is positively correlated with the catchment area and the length of the flow path in the upstream watershed of the depression, and negatively correlated with the main channel slope.

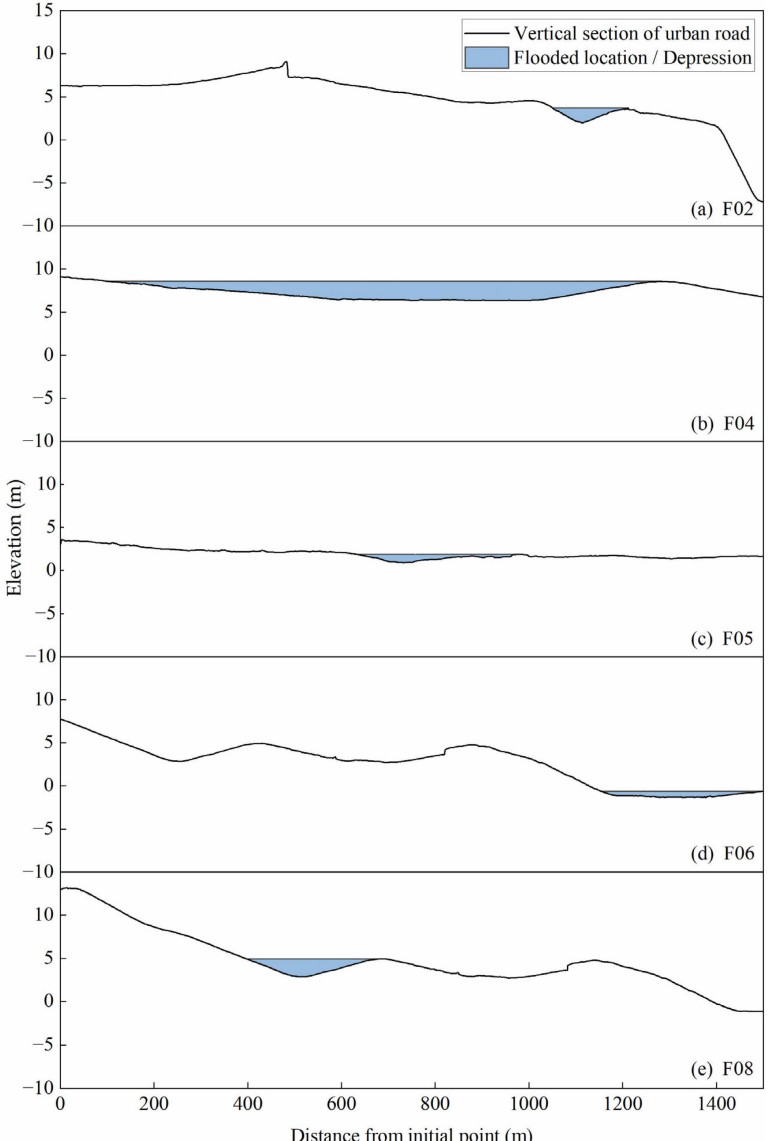

**Figure 4.** Vertical section of the local depressions where the flood-prone points are located. (**a**) Flood-prone point "F02"; (**b**) Flood-prone point "F04"; (**c**) Flood-prone point "F05"; (**d**) Flood-prone point "F06"; (**e**) Flood-prone point "F08".

**Table 4.** Flood characteristics on 26 March, 23 April and 7 June 2022 at the five flood-prone points.

| Number of Flood-Prone Points | | | F02 | F04 | F05 | F06 | F08 |
|---|---|---|---|---|---|---|---|
| Flood characteristics | MFD (m) | (1) 23 April | 1.05 | 0.28 | 0.31 | 0.35 | 0.57 |
| | | (2) 7 June | 0.33 | 0.19 | 0.16 | 0.19 | 0.13 |
| | Variation of the MFD (m) | (1)–(2) | 0.72 | 0.09 | 0.15 | 0.16 | 0.44 |
| | FPLT (min) | 26 March | 10.00 | 1.23 | 2.45 | 0.00 | 0.00 |
| | | 23 April | 18.00 | 9.00 | 11.47 | 19.52 | 8.23 |
| | | 7 June | 21.00 | 13.23 | 17.23 | 11 | 13.23 |
| Terrain parameter | Depression | B/H | 93.04 | 532.28 | 368.58 | 1466.84 | 143.02 |
| | Catchment | A (km²) | 1.711 | 0.109 | 0.130 | 0.179 | 0.048 |
| | | L (m) | 2814.36 | 674.02 | 693.16 | 477.77 | 442.41 |
| | | G | 0.009 | 0.012 | 0.006 | 0.018 | 0.023 |

Note(s): [6]: "MFD" is defined as maximum flood depth, "FPLT" is defined as flood peak lag time, "B/H" is defined as the ratio of the maximum probable inundation length (B) to the water depth (H), "A" is defined as the catchment area, "L" is defined as flow path length, and "G" is defined as the flow path gradient.

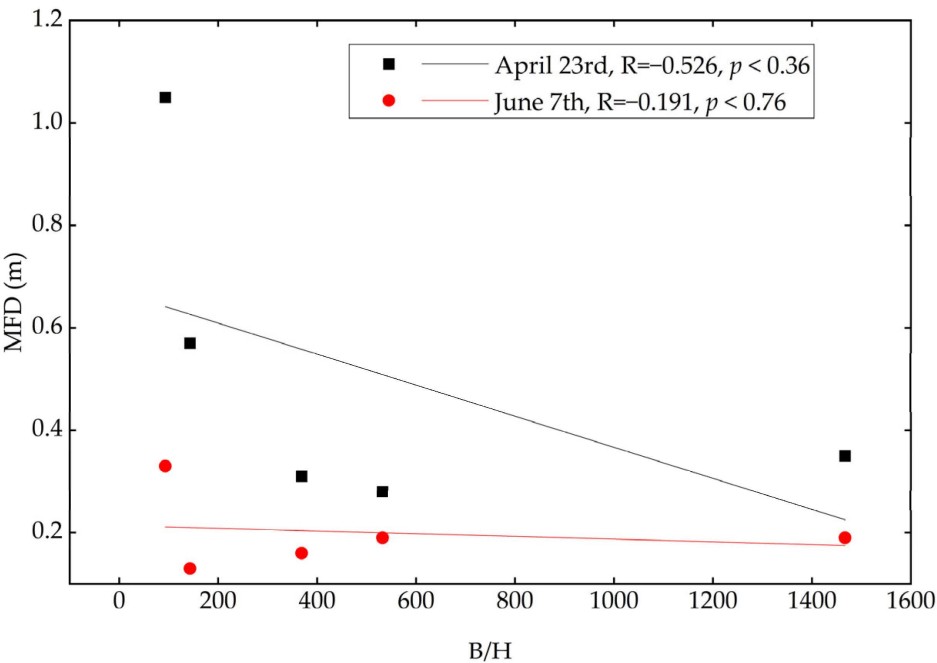

**Figure 5.** Correlation between maximum flood depth (MFD) and the ratio of the maximum probable inundation length to the water depth (B/H).

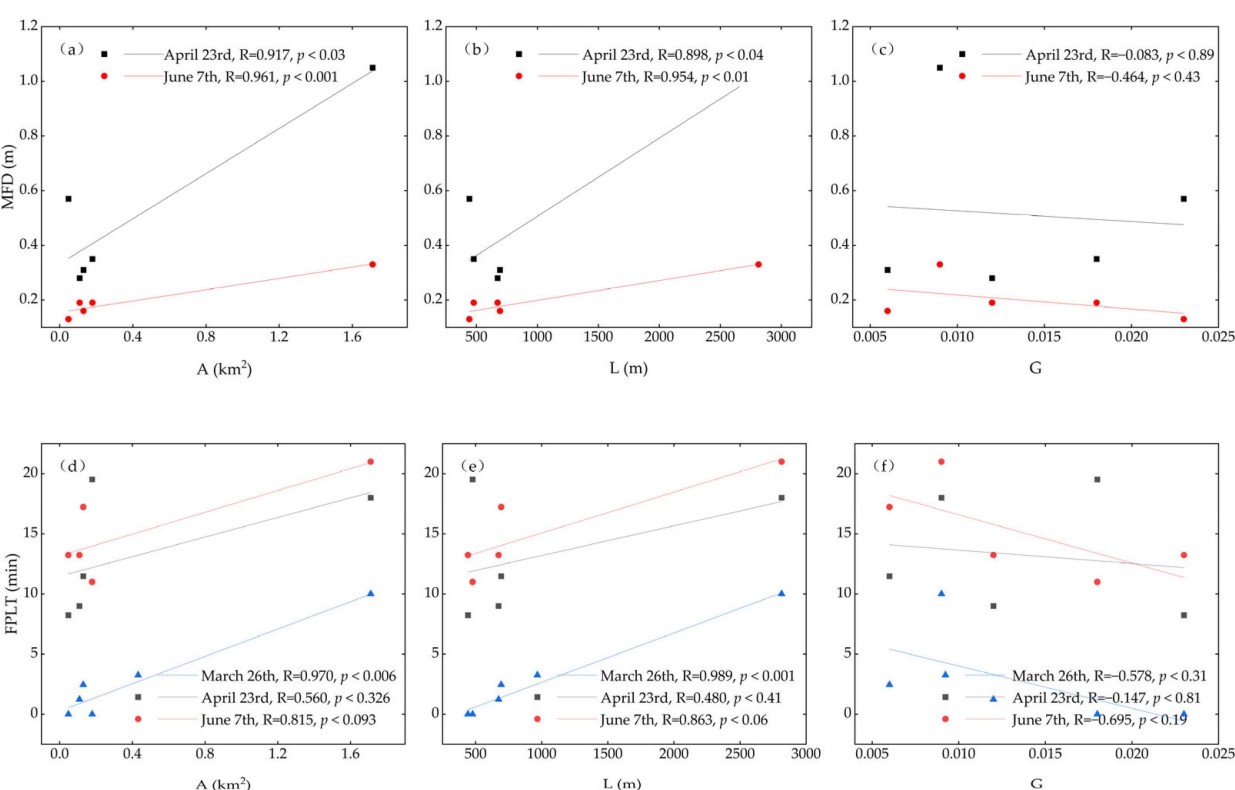

**Figure 6.** Correlation between flood characteristic parameters (MFD and flood peak lag time(FPLT)) and terrain parameters of the catchment (catchment area (A), flow path length (L), and flow path gradient (G),). (**a**) Correlation between the MFD and A. (**b**) Correlation between the MFD and L. (**c**) Correlation between the MFD and G. (**d**) Correlation between the FPLT and A. (**e**) Correlation between the FPLT and L. (**f**) Correlation between the FPLT and G.

The underlying surface of a mature city often remains steady over a period. Therefore, terrain parameters generated from the underlying surface data should similarly remain largely stable. Therefore, it can be inferred that the impact of topography on urban surface flooding will also remain stable for a long time. This stability will give rise to hope for creating a dependable and rational plan to control urban flooding.

## 4. Discussion

### 4.1. Relationship between the Distribution of Flood-Prone Points of Urban Roads and Terrain

This research aimed to evaluate the influence of terrain on the surface flood characteristics of urban roads. To achieve this, this study employed spatial interpolation techniques to reconstruct road terrain parameters and applied statistical analysis to investigate the quantitative relationship between terrain and surface flooding. The analysis of spatial correspondence between flood-prone locations and urban topography revealed a strong spatial association, indicating that flood-prone areas consistently occur in depressed regions on road surfaces. This conclusion can be easily explained. Rainwater on urban roads flows naturally downhill due to gravity. When there are local depressions, and the drainage system is insufficient, the rainwater collects in these depressions and flows downstream along the road surface, leading to localized flooding. Without depressions, water cannot accumulate and instead continues to flow downwards. Therefore, flood-prone areas on urban roads are typically found in depressions, which are critical for the risk analysis of road flooding. However, as depicted in Figure 2, not all depressions (characterized by negative curvature) develop into flood-prone points. The formation of flood-prone points also depends on the capacity of the drainage system to absorb rainwater. If the drainage capacity is sufficient, surface runoff can be effectively directed into rivers through drainage systems or absorbed by sponge facilities, preventing urban flooding [23,74]. Hence, depressions serve as a necessary but insufficient condition for forming flood-prone points on urban roads. They should be given particular attention during urban flood risk analysis.

The analysis reveals a significant negative correlation ($p < 0.05$) between the distribution of flood-prone points and terrain curvature, indicating that areas with greater surface depression are more likely to be affected by floods. Using spatial interpolation to obtain curvature parameters provides an effective measure for describing surface depressions [39,54], which can be valuable for future applications in urban flood simulations. However, it is essential to carefully select the appropriate sampling scale when calculating curvature to ensure that the generated topographic parameters accurately reflect the actual terrain changes. Deviating from the optimal sampling scale by choosing a smaller or larger scale than the road width may result in an incomplete representation of the true topography.

The study area was divided into two terrain types: LL/GS and HL/SS. Analysis of variance (ANOVA) revealed significant variations in elevation and slope between these terrain types, indicating distinct terrain characteristics. However, no significant differences were observed in curvature parameters, suggesting that the distribution of surface depressions (curvature < 0) on urban roads is not influenced by elevation or slope changes. These depressions appeared to be randomly distributed throughout the road network. The ratio of NFPS/TNS showed minor differences between the two terrain types, indicating that the distribution of flood-prone points is not significantly influenced by elevation or slope. The study area represents a small-scale catchment where flood-prone points are concentrated within a relatively small area, suggesting similar rainfall conditions. After eliminating the influence of rainfall conditions and terrain factors, the significant differences observed in the correlation analysis between different terrain categories can be solely attributed to variations in the efficiency and functionality of the drainage systems. In previous pipe network designs, the economic cost is a crucial consideration, and the desired slope of drainage pipelines is typically designed to closely match the natural slope of the surface [75,76]. In the LL/GS areas, which have a low surface elevation and gentle overall slope, the slope at the base of the drainage pipeline is likely to be relatively gradual. Additionally, as shown in Figure 3, this area is closer to the city's natural water system and represents

the downstream region, occasionally affected by outlet water pressure. Additionally, it experiences higher overall drainage pressure compared to the HL/SS areas. Thus, it can be inferred that the impact of terrain on the distribution of flood-prone points is limited by the performance of the drainage system. This finding aligns with previous studies by Komori et al. [77] and Rahmati et al. [78], suggesting that while terrain characteristics significantly influence the formation of urban flood-prone points, they are not the sole determining factor. Human-made factors, such as the drainage system's performance, are considered more critical.

### 4.2. Influence of Terrain on Flood Characteristics at Flood-Prone Points

Although the terrain does not entirely determine the formation and distribution of flood-prone points, it has significant effects on the flooding characteristics in these points. They determine the maximum depth, submerged range, and maximum ponding volume of the flooding process at each flood-prone point, and it also affects the variation characteristics of MFD under different rainfall conditions and potential emergency response time (FPLT). Specifically, a larger upstream catchment area and longer flow paths contribute to greater peak flood depths and longer emergency response times. In contrast, the influence of flow path gradient on flood risk is more pronounced and unequivocal. A larger flow path gradient undoubtedly amplifies the potential for greater flood risk, characterized by increased flood depths and shorter response times. By including these terrain-related parameters and their influence on flood occurrence in urban flood defense manuals, stakeholders can better understand the hazards associated with flooding and implement necessary measures based on this clear and comprehensible information.

### 4.3. Potential Strategies for the Study Area

Researching the mechanisms of urban flood disasters is crucial for guiding scientific prevention and control strategies. Effective disaster management and preventive strategies can greatly reduce the likelihood of flooding [79,80]. According to the findings of this study, there is a significant association between the formation of flood-prone points and the presence of depressions. In areas with poor drainage, larger depression depths are associated with higher flood risk. Hence, urban flood risk analysis should prioritize areas characterized by subsidence [81]. Effectively describing depressions helps to improve the accuracy and efficiency of flood simulations [12,82]. The use of spatial interpolation methods to compute the terrain curvature parameters in this study provides an efficient means of characterizing the depth of depressions.

The findings of this study emphasize the crucial role of drainage system performance in urban road surface flooding. Enhancing the performance of drainage systems, through measures like adjusting inlet shape and size [29], pipe repair [83], and effective maintenance planning [84,85], is vital for improving urban flood resilience. Additionally, it is essential to consider the influence of depression morphology and upstream catchment parameters on flood characteristics during urban drainage planning. Implementing appropriate designs for surface depression morphology and upstream catchment areas can effectively reduce systemic flooding risk in these regions [86–89].

Another significant finding of this study relates to the management of flood risks. A critical aspect of flood prevention and mitigation is the effective communication of specialized hydrological information to the general public [90], ensuring their comprehensive understanding and retention of this information [90,91]. By educating individuals on proactive measures to avoid disasters, the effectiveness of mitigation activities can be greatly improved [92]. This research demonstrates that despite the inherent uncertainty of urban flood disasters [11–13], flood-prone areas consistently exhibit certain characteristics influenced by terrain constraints. These characteristics include the maximum inundation extent, water depth, volume, and potential emergency response times. Importantly, it should be noted that the areas with the highest risk of significant flood depths may not necessarily coincide with regions experiencing frequent flood events. The provision of this

clear, concise, easily comprehensible, and specific information is expected to play a pivotal role in flood disaster management [91,93].

## 5. Conclusions

Roads serve as primary conduits for urban drainage and are highly vulnerable to flooding, especially in low-lying areas such as underpasses, tunnels, and culverts. Understanding the impact of terrain on surface flood characteristics in urban road systems is crucial for developing effective flood prevention strategies. However, current terrain data obtained from surveys or digital elevation models (DEMs) have limitations. This study employed spatial interpolation methods to assign measured elevation points to the road network. Using statistical analysis, we investigated the relationship between terrain parameters (elevation, slope, and curvature) and flood characteristics. The key findings are as follows:

(1) The distribution of flood-prone points in the urban road network shows a strong spatial correspondence with the curvature parameters of the terrain. With an appropriate sampling scale, the coincidence rate can even reach 100%. Depressions serve as a necessary but insufficient condition for forming flood-prone points on urban roads.

(2) In LL/GS areas characterized by higher drainage pressure, there is an extremely significant negative correlation between the distribution of flood-prone points and terrain curvature (Spearman's r = 0.205, $p < 0.01$). However, in HL/SS areas, no significant correlation was observed between them.

(3) Terrain significantly influences the distribution of flood-prone points, but drainage system performance is crucial and outweighs terrain factors.

(4) The terrain of depressions plays a significant role in determining the maximum depth of flood, submerged area, and ponding volume during urban pluvial flooding.

(5) The morphological characteristics of the upstream catchment area within the depressions also affect the MFD and the FPLT. Larger upstream catchment areas and longer flow paths typically result in greater MFD and longer FPLT (emergency response times). Moreover, a steeper flow path gradient directly contributes to higher flood risk (greater MFD and shorter FPLT).

Understanding urban flood mechanisms is vital for effective disaster management. The results of our study can inform flood risk identification, guide the formulation of structural and non-structural flood control measures, and mitigate decision-making risks associated with uncertainties.

**Author Contributions:** Conceptualization, X.Z.; methodology, X.Z.; software, Q.S.; validation, M.Y. and A.K.; formal analysis, X.Z.; investigation, X.Z.; resources, A.K.; data curation, A.K.; writing—original draft preparation, X.Z.; writing—review and editing, A.K.; visualization, Q.S.; supervision, A.K.; project administration, X.L.; funding acquisition, H.W. All authors have read and agreed to the published version of the manuscript.

**Funding:** This research was funded by the Young Scientists Fund of the National Natural Science Foundation of China, grant number 51809283, and the National Key Research and Development (R&D) Program Project, grant number 2018YFC0407902.

**Data Availability Statement:** Not applicable.

**Acknowledgments:** Thanks to the assistance provided by the Bureau of Water Resources of Guangzhou Municipality.

**Conflicts of Interest:** The authors declare no conflict of interest.

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
