# Peer review of "Influence of Terrain Factors on Urban Pluvial Flooding Characteristics: A Case Study of a Small Watershed in Guangzhou, China"

_water, doi:10.3390/w15122261_

Round 1

Reviewer 1 Report

///

The authors presented an interesting and highly useful paper on assessing the influence of terrain characteristic factors on urban fluvial flooding in a small watershed. The underlying concepts are clearly presented and well explained. The reviewer has not found any major weaknesses of the paper, and his/her assessment has focused on improving the paper understanding and clarity. By implementing minor comments listed below.

Minor comments:

Line 16 – I would add also underpasses as locations, where flooding/inundation starts early during a storm

L39 – I would replace “rainwater facility drainage system” by storm sewer drainage system

Lines 50-56 – somewhere here one should introduce the terminology of a minor and major drainage system: the minor system – small drainage elements conveying frequent events (e.g., storm sewer pipes, events with return periods up to 5-25 years), the major drainage system – drainage elements conveying larger flows, with return period up to 50-100 years (e.g., roads, urban creek channels). 

L113- 114 – awkward wording, please change

L128 – text – sampling of what?

L34 – “artificially” designed – meanings?

135 – incomplete sentence.

159 – as high as -0.05-0.05 – meaning?

L290 – check the wording.

L330-331 – check the grammar

L348, L349 – check the text

L398 – instead of fewer, use less.

The authors presented an interesting and highly useful paper on assessing the influence of terrain characteristic factors on urban fluvial flooding in a small watershed. The underlying concepts are clearly presented and well explained. The reviewer has not found any major weaknesses of the paper, and his/her assessment has focused on improving the paper understanding and clarity. By implementing minor comments listed below.

Minor comments:

Line 16 – I would add also underpasses as locations, where flooding/inundation starts early during a storm

L39 – I would replace “rainwater facility drainage system” by storm sewer drainage system

Lines 50-56 – somewhere here one should introduce the terminology of a minor and major drainage system: the minor system – small drainage elements conveying frequent events (e.g., storm sewer pipes, events with return periods up to 5-25 years), the major drainage system – drainage elements conveying larger flows, with return period up to 50-100 years (e.g., roads, urban creek channels). 

L113- 114 – awkward wording, please change

L128 – text – sampling of what?

L34 – “artificially” designed – meanings?

135 – incomplete sentence.

159 – as high as -0.05-0.05 – meaning?

L290 – check the wording.

L330-331 – check the grammar

L348, L349 – check the text

L398 – instead of fewer, use less.

Author Response

请参阅附件。

Reviewer 2 Report

The paper shows a very interesting idea for solving the issue of terrain properties of flooding. A literature review is fine; the methodology goes toward solving the problem but is unsatisfactory. 

I know that the authors were using a lot of their time and energy to produce the paper, but I regret that I am suggesting rejecting it.

-authors enclose superficial maps/profiles of the terrain. How could actual research and conclusions be done only by analyzing Figure 4 and previous tables? Any information about the porosity or texture of the soil? 

-conclusions were done on the basis of picture 5. Such could not be provided on the presented values, grouped only on two positions in the diagram. There is a too small amount of observed data, i.e., too small values for any accurate conclusions. 

-authors didn't enclose a larger map of the location, i.e., where the area of Guangzhou is in the map of China.

-authors should avoid writing in the first face. For example, instead of ''we have'', there should be ''it has'' or something similar. The writing style should be neutral. 

-entire text should be checked by the English language professor or even the native English speaker. 

-entire text should be checked by the English language professor or even the native English speaker. There is too much errors there. 

Reviewer 3 Report

Dear authors, it was a pleasure to review your manuscript "Study on the influence of terrain factors on urban pluvial flooding characteristics: a case study of a small watershed in Guangzhou, China". I think you have done a good research, the methodology is good as well as the presentation of the results. There are just a few suggestions for you that you can see below.

I suggest you make a better version of Fig. 5, in this form its parts are scattered and separated.

I think the conclusion of your research should be better. The advantages and novelties resulting from your research should be particularly emphasize.

Best regards.

Round 2

Reviewer 2 Report

The authors made specific changes in the manuscript. I have carefully read all their comments in the answers to each review. 

Still, the required changes do not satisfy demands from my side. 
